# Trends in Hospitalization and Economic Impact of Percutaneous Kyphoplasty in Italy

**DOI:** 10.3390/jcm11247464

**Published:** 2022-12-16

**Authors:** Umile Giuseppe Longo, Rocco Papalia, Luca Denaro, Sergio De Salvatore, Laura Ruzzini, Ilaria Piergentili, Vincenzo Denaro

**Affiliations:** 1Research Unit of Orthopaedic and Trauma Surgery, Fondazione Policlinico Universitario Campus Bio-Medico, Via Alvaro del Portillo, 00128 Roma, Italy; 2Research Unit of Orthopaedic and Trauma Surgery, Department of Medicine and Surgery, Università Campus Bio-Medico di Roma, Via Alvaro del Portillo, 00128 Roma, Italy; 3Academic Neurosurgery, Department of Neurosciences, University of Padua, 35128 Padua, Italy; 4Department of Surgery, Orthopedic Unit, Bambino Gesù Children’s Hospital, 00050 Rome, Italy

**Keywords:** kyphoplasty, vertebroplasty, vertebral fractures, percutaneous, epidemiology

## Abstract

Vertebral Fractures (VFs) caused by osteoporosis are the most typical reason for performing Percutaneous Kyphoplasty (PK). Globally, VF prevalence is not well described in the literature. In Europe, only Sweden has an accurate record of the incidence of this type of fracture. Moreover, the exact incidence of the PK procedure is not reported. Therefore, the annual patterns and financial burden of PK in Europe is not well known, and it could be may better understood by examining national registers. For surgeons, lawmakers, hospital administrators, and the healthcare system, determining the annual national costs of this treatment is helpful. The National Hospital Discharge Reports (SDOs) submitted to the Italian Ministry of Health during the years covered by this study (2009–2015) were used. A total of 13,113 kyphoplasties were performed in Italy, with a prevalence of 3.6 procedures for every 100,000 Italian inhabitants over 15 years. The average age of patients was 68.28 (±12.9). Females represented the majority of patients undergoing PK procedures (68.6%). The median length of hospital stay was 5.33 days. The prevalence of PK procedures increased from 2009 to 2015, while the average days of hospitalization decreased. Older patients were most at risk in undergoing PK procedures. Reporting the national data on PK in Italy could also help compare the findings across nations. The current study aims to determine the trends of PK hospitalisation and patient features in Italy.

## 1. Introduction

The best treatment strategy for symptomatic Vertebral Fracture (VF) is a combination of rest, painkillers, muscle relaxants and bracing [1]. The majority of patients respond well to conservative treatment, but in nonresponsive cases, spinal augmentation is recommended [2,3]. 

Percutaneous kyphoplasty (PK) is a surgical procedure for the treatment of VF, developed in 1998 as a variation of the percutaneous vertebroplasty (PV) technique [1]. PK could restore vertebral height and kyphotic spine angulation, reduce pain, and give the patient earlier mobilization [4]. Compared to PV, PK reports a lower cement leakage, better short-term pain relief, and kyphotic angle restoration [5]. However, PK requires lengthier operation times and higher costs than PV [6]. In addition, the success rate is influenced by the type of fracture, the localization, and the severity of the injury [7]. 

VFs caused by osteoporosis are the most typical reason for performing PK [8]. Osteoporosis annually leads to 750,000 VFs in the United States [9]. However, because PK is less frequently used than PV, the precise incidence of PK worldwide has not been thoroughly analysed in the literature [1,2]. In Europe, only Sweden has an accurate record of the incidence of this type of fracture. However, the exact incidence of the PK procedure is not reported [10]. 

The available evidence does not support the regular use of vertebral augmentation to reduce pain in the VF [11], and additional high-quality clinical trials are needed to produce accurate data. To prevent treatment failure or adverse consequences, a correct surgical indication is required. However, there has been no international agreement on the appropriate surgical indication for PK [12,13].

The mean estimated cost of PK (including hospital stay and materials) is EUR 7512.53, with an average cost to the patient of € 7610.97. The estimated profit for the hospital is about EUR 98.44 per procedure [14]. Multiplying these data for a large number of procedures, it is evident that the economic impact of PK is significant for the health care system. However, the exact cost depends on the healthcare system and the country.

Considering the controversial utility of this procedure, the lack of an accurate incidence could underestimate the economic burden of PK. Therefore, the annual patterns and financial burden of PK in Europe may be better understood by examining national registers. For surgeons, lawmakers, hospital administrators, and the healthcare system, determining the annual national costs of this treatment is helpful. 

Reporting the national data on PK in Italy could also help compare the findings across nations. The current study used the Nationwide Hospital Discharge Reports (SDOs) database to gather information on a national cohort of patients hospitalized for PK between 2009 and 2015. The current study aims to determine the trends of PK hospitalisation and patient features in Italy.

## 2. Materials and Methods

The National Hospital Discharge Reports (SDOs) submitted to the Italian Ministry of Health from 2009 to 2015 were used. These reports offered information on each hospital admission in Italy from public and private institutions. In Italy, it is the responsibility of the regional authorities to plan and monitor healthcare services provided through public or private structures. Therefore, hospitals regularly compile information on healthcare services and send them to the Ministry of Health [15]. These anonymous statistics give information about the patient’s age, sex, number of days spent in the hospital, diagnoses, and treatments [16]. In addition, population data were obtained from the National Institute for Statistics (ISTAT) each year [17].

The International Classification of Diseases, Ninth Revision, Clinical Modification (ICD-9-CM), procedure code: 81.66, was used to define PK. The study was focused on the Italian community over the age of 15. There were only 8 PKs for the class of patients aged 0 to 14 throughout the 7-year study period. Therefore, we excluded them, avoiding underestimating the population who underwent PK.

### Statistics

Descriptive statistical analyses were used to determine the annual number of PKs, the proportion of males and females, the average age, the number of days spent in the hospital, and the primary diagnosis. For categorical data, frequencies and percentages were used. For continuous variables, means and standard deviations were adopted. The annual adult population size from ISTAT was used to determine prevalence rates. The prevalence was calculated using data from the total Italian population over the age of 15. The data analysis was made using SPSS version 26 (Statistical Package for Social Sciences). Graphs were created using Microsoft Excel software (Version 16.67 (22111300)).

## 3. Results

### 3.1. Population

A total of 13,113 PK hospitalizations were reported in Italy during the 7-year study period, with an incidence of 3.6 procedures per 100,000 Italians over the age of 15. The prevalence of surgeries increased from 2.0 to 3.9 per 100,000 people between 2009 and 2015 (Figure 1). 

The average patient age from 2009 to 2015 was 68.28 (12.9). The age group of 75 to 79 years had the highest prevalence of procedures during the study period (Figure 2). 

Throughout the overall period, women had increased ages compared to men (Figure 3). 

Figure 4 shows the number of PKs in Italy by age group and gender during the study period. Both overall and throughout time, the majority of patients undergoing PK operations were female (female 68.6% and male 31.4%). In addition, male patients represented a larger portion of the patient population between the ages of 15 and 49, while female patients began to predominate after age 50.

### 3.2. Number of Days Spent in the Hospital

The average hospital stay was 5.3 days (1–110 days). Figure 5 shows the trend in decreasing the average number of days spent in the hospital. On average, males stayed in the hospital for more days than females (males 6.2 days and females 4.9). Younger patients had more days of hospitalization on average. 

### 3.3. Admission Diagnosis Codes

During the 7-year study period, the main primary diagnoses were closed fracture of the lumbar vertebra without mention of spinal cord injury (32.4%), pathologic fracture of vertebrae (30.0%), closed fracture of dorsal (thoracic] vertebra without mention of spinal cord injury (15.8%), osteoporosis, unspecified (4.3%), traumatic spondylopathy (2.9%), and closed fracture of unspecified vertebral column without mention of spinal cord injury (2.7%) (Figure 6).

### 3.4. Economic Impact

For each PK hospital admission, the actual mean Italian hospital reimbursement ranges from EUR 2296 (1-day procedure) to EUR 4629 (more than 1-day stay, with an increment of EUR 13 for every additional day of hospitalisation). From 2009 to 2015, there has been an estimated overall cost of EUR 56,352,355 (annual mean EUR 8,050,336 ± EUR 1,708,783; range from EUR 4,474,545 in 2009 to EUR 9,358,635 in 2012) for PK procedures in Italy.

## 4. Discussion

This research aims to determine the burden of PK hospitalizations in Italy. According to the results, there were 3.6 cases of PK per 100,000 people from 2009 to 2015. Over the years of the study, the reported number of procedures doubled, rising from 2.0 (per 100,000 people) in 2009 to 3.9 in 2015. A total of 68.6% of the patients were female of 75–79 years group. Male patients had more extended hospitalisations than female patients in the under 50 group. After 50 years, this ratio turns, probably as a result of postmenopausal osteoporosis, which raises the incidence of osteoporotic VFs [18]. 

The two most frequent diagnosis codes were “pathologic fracture of vertebrae” (30.0%) and “closed fracture of the lumbar vertebra without mention of spinal cord injury” (32.4%). In contrast, just 4.3% of cases were reported to have “osteoporosis, unspecified” as a diagnosis. This information conflicts with the literature, which claims that osteoporosis is the primary cause of VFs [10,19,20].

The number of days spent in the hospital decreased over time. Compared to women, men reported an average of more days in the hospital. Additionally, individuals (mainly men) between the ages of 25 and 29 reported more hospitalization days than the other groups. This hospitalisation duration is in line with results from the USA, where the mean length of stay varies between 5.5 [21] and 4.4 days [22]. This information may seem surprising because PK is typically a day procedure. Comorbidities and complications in the patients may contribute to more extended hospital stays. The ICD-9 does not reflect the postoperative issues that may impact the length of stay. As a result, the reasons for the prolonged hospital stays were not reported.

About ten million people worldwide are affected by osteoporosis, which also accounts for more than 700,000 VFs and 150,000 hospitalizations annually in the US [10]. Infections and tumours are the secondary causes of non-traumatic VFs [9,10]. Osteoporosis, steroid use, menopause, hyperthyroidism, chronic illness, and kidney disease are the most typical risk factors for VFs [23]. Osteoporotic VFs typically advance slowly and without symptoms. 

Increased osteoporotic fractures and PK procedures are directly related to the population’s ageing and longer life expectancy [20].

Osteoporotic fractures often proceed slowly and subclinically; however, patients may experience severe pain and a reduced range of motion (ROM) in acute collapse. Rest, NSAIDs, muscle relaxants, bracing, and physical therapy are all components of the traditional, conservative approach to managing VFs [24,25]. The use of NSAIDs, however, was debatable [25]. PK is recommended when pain persists for more than two weeks after receiving conservative treatment where there is chronic immobility, painful spinal osteonecrosis, or where there are painful metastases or painful vertebral hemangiomas [26,27].

PV and PK have the same surgical indications and complications [28,29]. PK may change the vertebral segments’ biomechanical properties, raising the risk of adjacent compression fractures [30]. Otherwise, these data are still debated. According to Taylor et al. [31], patients treated with PK experienced lower prevalence in the adjacent VF than those treated with PV.

As reported in the results, the economic burden of PK is significant in Italy. However, compared to US, according to the study by Laratta et al., this burden is lower [32]. The authors reported that the national cost for PK between 2008 to 2014 was about USD 8,849,590,697.

However, several studies in the literature have reported controversial results about the efficacy of PK in osteoporotic VF [33]. Therefore, further high quality randomized clinical trials are mandatory to assess the real efficacy of PK, as they could change the clinical practice in patients with VF. In 2009 this was undertaken with randomized controlled trials by Buchbinder and Kallmes regarding the efficacy and safety of vertebroplasty [19,34]. The results of the study led to a change in the number of vertebroplasties worldwide. 

### Limitations

The administrative data used in the current study were obtained from several hospitals and macro-regions. All reported procedures followed the International Classification of Diseases 9 (ICD-9) guidelines. Otherwise, several ICD-9 codes could be used for the same surgical procedure. Therefore, the variability of codification could cause our results to be overestimated. Similarly, the diagnosis codes of ICD-9 are general; therefore, it was impossible to report the exact diagnosis of the procedures performed.

Furthermore, this paper was a non-comparative epidemiological study. This article aimed to report data on PK procedures in Italy and to compare them with other countries. Other studies already performed the same aim regarding PV [35]; therefore, data on this technique were not reported. Lastly, with SDO reports, it was impossible to assess the radiological parameters and the outcomes of the patients; therefore, no clinical comparison in the population of this study could be provided. Moreover, the number of lumbar VFs was higher compared to thoracic ones. Usually, this relationship is inverse. Unfortunately, due to the limits of the ICD-9 system it was not possible to assess and check the single levels; further studies are therefore required to clarify this point. 

Furthermore, the data of this study seem to be not recent, as they cover a period from 2009 to 2015. However, despite the regional institution providing the statistics annually to the Ministry of Health, the latter does not release the data regularly. Therefore, the results of the present study represent the most up-to-date epidemiological data on PKs performed in Italy.

## 5. Conclusions

The prevalence of PK procedures increased from 2009 to 2015, while the average days of hospitalization decreased. Older patients were most at risk of undergoing PK procedures. It is necessary to perform further clinical trials to define the proper indication of this procedure. Epidemiological studies allow us to examine the national variation in specific surgical procedures and compare these with other countries, allowing researchers to compare outcomes and define surgical indications.

## Figures and Tables

**Figure 1 jcm-11-07464-f001:**
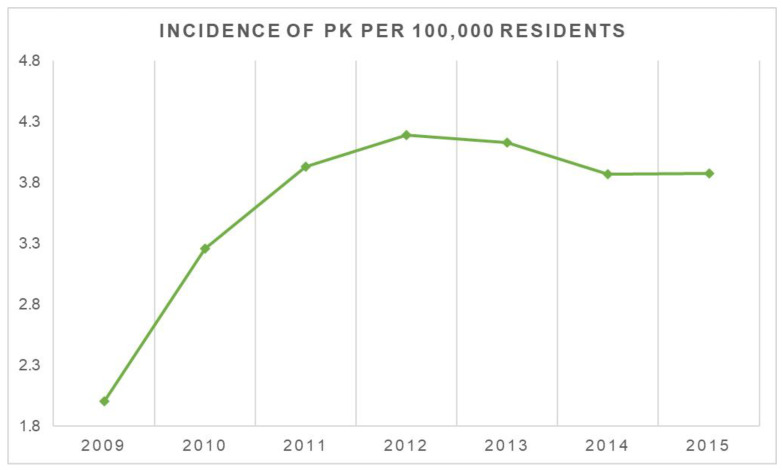
Incidence of PK per 100,000 residents from 2009 to 2015.

**Figure 2 jcm-11-07464-f002:**
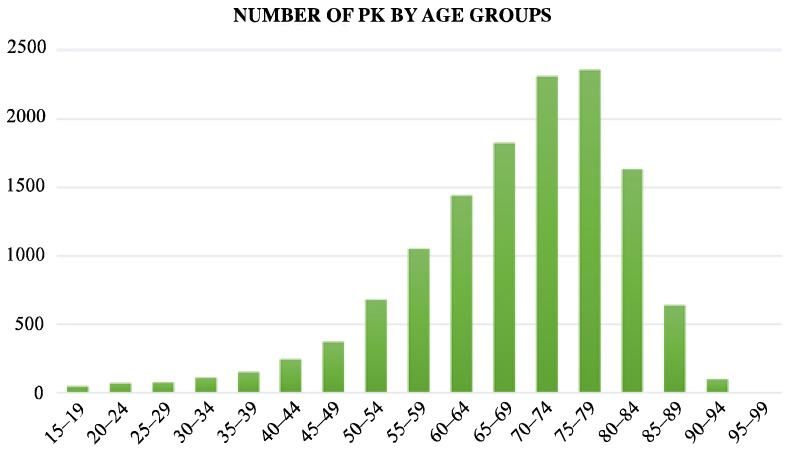
Number of PK by age group.

**Figure 3 jcm-11-07464-f003:**
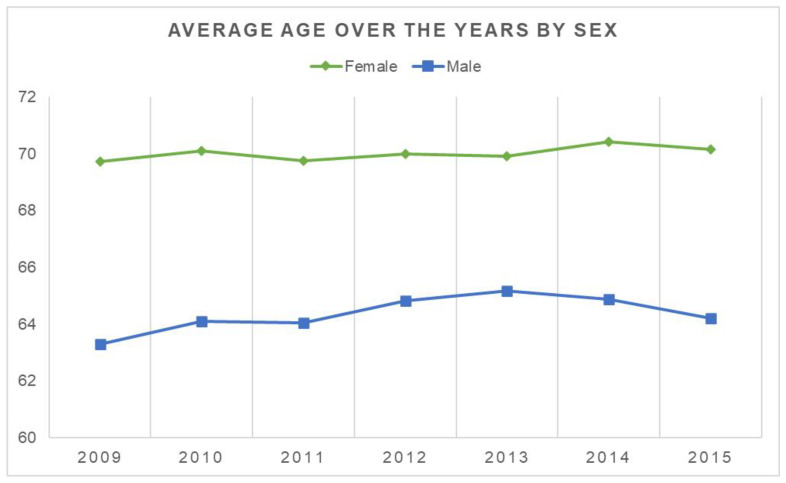
Mean age of PK divided by sex.

**Figure 4 jcm-11-07464-f004:**
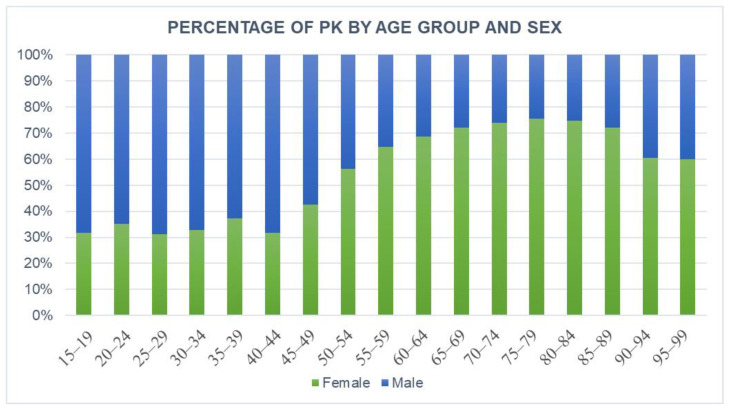
Percentage of PK divided by age group and sex.

**Figure 5 jcm-11-07464-f005:**
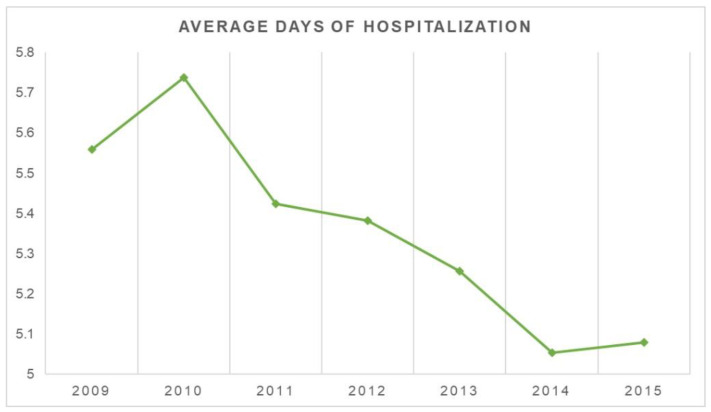
Average days of hospitalization.

**Figure 6 jcm-11-07464-f006:**
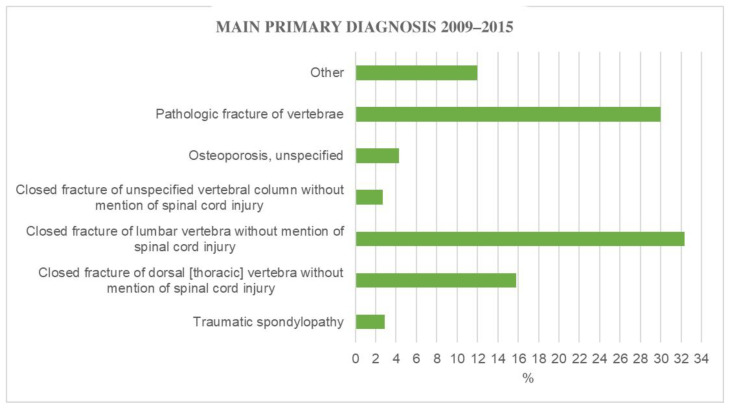
Main primary diagnosis for PK from 2009 to 2015.

## Data Availability

The datasets analysed during the current study are not publicly available but are available from the corresponding author on reasonable request.

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
