# Peer review of "Trends in Hospitalization and Economic Impact of Percutaneous Kyphoplasty in Italy"

_jcm, 2022, doi:10.3390/jcm11247464_

Round 1

Reviewer 1 Report

The current study reported the trends of percutaneous kyphoplasty(PKP) hospitalisation and patient features from 2009 to 2015 in Italy. The results showed the prevalence of PK procedures increased from 2009 to 2015, while the average days of hospitalization decreased, and the older patients were more at risk of PK procedures.

 Though the paper was well written, the data included in this study was from 2009 to 2015, which was not updated, and the finding bring little new information.

Author Response

We are honoured you appreciated our paper. the changes performed were highlighted in yellow in the manuscript.

We would thank you for giving us the possibility to explain this point. we know that the study's data seems to be not recent as they cover a period from 2009 to 2015. However, despite the regional institution providing the statistics annually to the Ministry of Health, the latter does not release the data regularly. Therefore, the results of the present study represent the most updated epidemiological data on PK performed in Italy.

we improved the limitation section accordingly.

Reviewer 2 Report

It is interesting that there were significantly more lumbar VFs than thoracic VFs. Usually the relationship is inverse. It would have been nice to see the level by level breakdown; however, I realize that the specifics of the available data are limited

Author Response

we want to thank you for the appreciation of our paper. We strongly agree with you, but as you mentioned, the data was limited and it was impossible to retrieve the information required from the SDO reports. unfortunately with the ICD-9 system, it is possible to perform coding mistakes, which could partially answer our questions. however, further clinical trials are required to obtain specific results. 

we improved the discussion section with your suggestions

Round 2

Reviewer 1 Report

I think the article meets the revision requirements